Lengthened partial repetitions elicit similar muscular adaptations as full range of motion repetitions during resistance training in trained individuals

Wolf Milo 1
Androulakis Korakakis Patroklos 1 polkarots@gmail.com
http://orcid.org/0000-0001-8186-1889 Piñero Alec 1
Mohan Adam E. 1
Hermann Tom 1
Augustin Francesca 1
Sapuppo Max 1
http://orcid.org/0009-0000-0723-2830 Lin Brian 1
Coleman Max 1
http://orcid.org/0000-0003-1201-2685 Burke Ryan 1
Nippard Jeff 2
Swinton Paul A. 3
http://orcid.org/0000-0003-4979-5783 Schoenfeld Brad J. 1
1 Department of Exercise Science and Recreation, Applied Muscle Development Laboratory, City University of New York, Herbert H. Lehman College , New York City , United States
2 STRCNG Incorporated OA Jeff Nippard Fitness , Oakville , Canada
3 School of Health, The Robert Gordon University , Aberdeen , United Kingdom
Yingling Vanessa
Electronic publication date: 2025 Feb 12
Publication date: 2025
Volume: 13
Electronic Location ID: e18904
Received 2024 Oct 2; Accepted 2025 Jan 6
Copyright: © 2025 Wolf et al.
Copyright year: 2025
Copyright holder: Wolf et al.
License: This is an open access article distributed under the terms of the Creative Commons Attribution License, which permits unrestricted use, distribution, reproduction and adaptation in any medium and for any purpose provided that it is properly attributed. For attribution, the original author(s), title, publication source (PeerJ) and either DOI or URL of the article must be cited.
License URL: https://creativecommons.org/licenses/by/4.0/

Keywords: Full rom, Lengthened partials, Hypertrophy, Range of motion, Long muscle lengths, Long length partials, Full range of motion, Muscle growth, Strength

Funding: STRCNG Incorporated o/a Jeff Nippard Fitness This study was supported with a grant from STRCNG Incorporated o/a Jeff Nippard Fitness. The funders had no role in study design, data collection and analysis, decision to publish, or preparation of the manuscript.

==============================
Purpose

Resistance training using different ranges of motion may produce varying effects on musclular adaptations. The purpose of this study was to compare the effects of lengthened partial repetitions (LPs) vs. full range of motion (ROM) resistance training (RT) on muscular adaptations.

Methods

In this within-participant study, thirty healthy, resistance-trained participants had their upper extremities randomly assigned to either a lengthened partial or full ROM condition; all other training variables were equivalent between limbs. The RT intervention was an 8-week program targeting upper-body musculature. Training consisted of two training sessions per week, with four exercises per session and four sets per exercise. Muscle hypertrophy of the elbow flexors and elbow extensors was evaluated using B-mode ultrasonography at 45% and 55% of humeral length. Muscle strength-endurance was assessed using a 10-repetition-maximum test on the lat pulldown exercise, both with a partial and full ROM. Data analysis employed a Bayesian framework with inferences made from posterior distributions and the strength of evidence for the existence of a difference through Bayes factors.

Results

Both muscle thickness and unilateral lat pulldown 10-repetition-maximum improvements were similar between the two conditions. Results were consistent across outcomes with point estimates close to zero, and Bayes factors (0.16 to 0.3) generally providing “moderate” support for the null hypothesis of equal improvement across interventions.

Conclusions

Trainees seeking to maximize muscle size should likely emphasize the stretched position, either by using a full ROM or LPs during upper-body resistance training. For muscle strength-endurance, our findings suggest that LPs and full ROM elicit similar adaptations.

Introduction

Resistance training (RT) is generally considered the most efficacious exercise modality for eliciting muscle hypertrophy in humans (Schoenfeld et al., 2021).

The effects of manipulating range of motion (ROM) during RT have been extensively studied, with many investigations focusing on training at longer muscle lengths (Wolf et al., 2023). While generally lacking ecological validity, five studies have compared longer vs. shorter muscle length isometric contraction (Akagi, Hinks & Power, 2020; Alegre et al., 2014; Hinks et al., 2021; Kubo et al., 2006; Noorkõiv, Nosaka & Blazevich, 2014), and nine studies have compared partial ROM at longer muscle lengths (referred to as lengthened partials and abbreviated as LPs) vs. shorter muscle lengths (referred to as shortened partials and abbreviated as SPs) on muscle hypertrophy (Kassiano et al., 2022a; Larsen et al., 2024a; Maeo et al., 2020, 2022; Mcmahon et al., 2014b; Pedrosa et al., 2022, 2023; Sato et al., 2021; Stasinaki et al., 2018)1 . Additionally, a recent study compared employing LPs following momentary failure using full ROM vs. full ROM alone and found that the former intervention resulted in greater muscle hypertrophy vs. full ROM (Larsen et al., 2024b)1. However, it is worth noting that the LPs group also completed a significantly higher total volume load, which may have influenced the hypertrophic outcomes.

More importantly, four studies have compared LPs to full ROM resistance training (Goto et al., 2019; Kassiano et al., 2022b; Pedrosa et al., 2022; Werkhausen et al., 2021). With the exception of Werkhausen et al. (2021), all remaining studies showed that training at longer muscle lengths elicited greater hypertrophy, suggesting that trainees aiming to maximize muscle growth should emphasize training at longer muscle lengths. Notably, the unique findings of Werkhausen et al. (2021) may be attributable to their use of a concentric-only protocol and the extreme limited range of motion (a 9° change in knee angle) used by the partials group. For strength, when comparing full ROM to LPs, previous studies have found that 1 RM increases are ROM specific (Pedrosa et al., 2022; Kassiano et al., 2022b).

Previous studies investigating manipulations to ROM have important limitations. Research on LPs vs. full ROM RT has used single-exercise interventions, which are not representative of typical RT routines that involve multiple exercises, limiting the ecological validity of these findings (Kassiano et al., 2022b). Additionally, most studies have focused on lower-body muscles (quadriceps, plantar flexors, hip extensors), with fewer studies on upper-body muscles (e.g.: elbow flexors and extensors) (Pedrosa et al., 2023; Goto et al., 2019). Moreover, these studies typically train muscles through the central 75% of a joint’s ROM, not at maximal muscle lengths, leaving it unclear if the benefits of longer muscle length training extend to the extremes of muscle length. Lastly, nearly all studies have been conducted on untrained individuals, limiting generalizability to trained populations.

In an effort to bridge gaps in the current literature, this study aimed to compare the effects of LPs and full ROM RT on upper-body muscle hypertrophy in resistance-trained participants using a multi-exercise routine, with a secondary aim to evaluate their effects on strength-endurance (i.e., 10 RM performance). Portions of this text were previously published as part of a preprint (https://doi.org/10.51224/SRXIV.455).

Materials and Methods

We opted for a within-participant design, assigning each limb to either LPs or full ROM resistance training, to compare their effects on upper-body muscle hypertrophy and strength-endurance over an 8-week, supervised multi-exercise program.

Participants

As with previous studies from our group (Burke et al., 2024), we adopted a Bayesian framework for our analyses with a focus on describing the most plausible values from our experiment vs. a dichotomous hypothesis testing of whether an effect existed or not. We adopted a within-participant design and included the use of informative priors to enhance precision. Anticipating a higher attrition rate due to the recruitment methods employed, we aimed to recruit thirty participants at the outset.

We estimated sample size based on expected precision of the average treatment effect using 95% credible intervals and simulated Bayes factor calibration. Informative priors, drawn from relevant meta-analyses (Wolf et al., 2023; Swinton et al., 2022), informed these simulations. Precision assessment across sample sizes of 20, 25, and 30 indicated sufficient reliability with a sample size of twenty to twenty-five participants (Table 1).

Table 1 Bayesian sample size determination assessing credible interval precision and simulation-based calibration of Bayes factors.

Sample size	95% Credible interval length for average treatment effect
[95%CrI]	Average posterior model probability
[95%CrI]	Average percentage of posterior allocated to H1 when H1 true	Average percentage of posterior allocated to H0 when H0 true	
N = 20	0.33
[0.27–0.39]	48.2
[38.5–58.0]	75%	76%	
N = 25	0.29
[0.24–0.34]	48.9
[40.1–56.0]	77%	79%	
N = 30	0.26 [0.22–0.30]	49.1
[41.8–55.7]	78%	80%	
Note:

Abbreviations: CrI, Credible interval.

Participants were admitted into the study based on the following criteria: (a) aged between 18–40 years; (b) free from existing cardiorespiratory or musculoskeletal disorders; (c) self-reported to have performed at least one upper-body resistance training session per week on more than 80% of weeks over the past 6 months; and (d) self-reported as free from the use of anabolic steroids or other illegal agents known to enhance muscle size currently and within the previous year. Participants were also instructed to refrain from consuming creatine products during the study period due to its potential impact on muscle growth when combined with RT (Burke et al., 2023).

Participants were recruited through participation in previous studies and the researchers’ personal networks, supplemented by social media posts. After meeting inclusion criteria, a mixed sample of participants was targeted, with emphasis on those with greater training experience. The methods of this study were preregistered prior to data collection and made publicly available on the Open Science Framework (https://osf.io/86v2h).

After being admitted to the study, participants’ upper limbs were randomly assigned to one of two experimental conditions: full range of motion (fROM; n = 30) or partial range of motion (pROM; n = 30) using counterbalanced block randomization with two limbs per block via online software (www.randomizer.org). Approval for the study was obtained from the City University Lehman College Institutional Review Board (#2024-0218). Written informed consent was obtained from all participants prior to beginning the study. All training and data collection were carried out at the same site.

To avoid potential dietary confounding of results, participants were advised to maintain their customary nutritional regimen as previously described (Burke et al., 2024). Dietary adherence was assessed by self-reported 5-day food records (including at least 1 weekend day) using MacroFactor (https://macrofactorapp.com/). Nutritional data were collected twice during the study, including 1 week before the first training session (i.e., baseline) and during the final week of the training protocol. Participants were instructed on how to properly record all food items and their respective portion sizes consumed for the designated period of interest. Each item of food was individually entered into the program, and the program provided relevant information as to total energy consumption, as well as the amount of energy derived from proteins, fats, and carbohydrates for each time-period analyzed.

Resistance training procedures

Participants completed two directly-supervised upper-body training sessions per week for 8 weeks, with at least one research assistant supervising each participant while performing the protocol outlined in Table 2.

Table 2 RT protocol.

Exercise	Sets	Repetition range	
Day 1			
Flat machine chest press	4	5–10	
Bench dumbbell row	4	10–15	
Dumbbell overhead triceps extensions	4	10–15	
Dumbbell supinating curl	4	10–15	
Day 2			
Incline machine chest press	4	10–15	
Cable single arm pulldown	4	5–10	
Cable pushdown	4	5–10	
Bayesian curl	4	5–10	

Participants were instructed to perform all sets to momentary muscular failure, defined as the point where they could no longer perform another repetition despite attempting to do so, with research assistants providing verbal encouragement and monitoring adherence to the prescribed ROM. The eccentric phase was performed in approximately 2 s, with a 1-s pause at the position where the target muscle was at its longest length. The concentric phase was executed with the intent to move the load explosively. Participants rested for 1 min when switching to the opposite limb between sets, alternating limbs set-by-set. The order of limb training was randomized and counterbalanced across sessions. Load adjustments were made as needed to maintain the target repetition range and intensity of effort on a set-to-set basis. To account for ecological validity, we included multiple repetition ranges across exercises. Training sessions were separated by a minimum of 48 h.

Participants were instructed not to perform any additional upper-body RT outside of the study protocol but were permitted to perform lower-body RT and other physical activities at their discretion. The research team included experienced researchers with PhDs in exercise-related disciplines as well as graduate-level students in human performance; a majority of the researchers and assistants also held certification in strength and conditioning (via the National Strength and Conditioning Association) and/or from nationally accredited personal training organizations. Training sessions were carried out between 9 a.m. and 5 p.m., with participants afforded the ability to train at the time of their convenience.

Range of motion: To achieve an ecologically valid operationalization for fROM and pROM protocols, participants received instruction from the research staff. To ensure the research staff provided standardized instructions to participants, videos were shown to both the research staff and participants, displaying the appropriate ROM and technique for each exercise (https://osf.io/a6cpz).

After randomization of the limbs and prior to the 8-week training program, participants underwent 10-repetition-maximum (10 RM) testing with fROM and pROM on both limbs in the unilateral lat pulldown exercise. fROM limb strength was always tested first, followed by pROM limb strength. The order of limb testing was randomized, counterbalanced, and standardized from pre- to post-intervention strength testing. Participants were instructed to perform the 10 RM testing following the ROM guidelines of their respective group. The 10 RM testing was consistent with recognized guidelines as established by the National Strength and Conditioning Association (NSCA, 2016).

After familiarization with full ROM, participants were instructed to complete a full ROM as comfortably as possible during training in the fROM condition. When the participant was unable to complete another full ROM repetition, the set was terminated. In the pROM condition, participants were instructed to perform half-repetitions (approximately 50% of full ROM), relative to their own individualized full ROM, from the position achieved at the end of the eccentric or lowering phase. The set was terminated when the participant attempted another partial ROM repetition with 50% of full ROM, but failed to complete the partial ROM repetition. The research staff provided instruction between and during sets as to whether the ROM achieved was adequate in both conditions.

Assessments

Participants underwent pre- and post-intervention testing in separate sessions, refraining from strenuous exercise for at least 72 h prior to testing. The following measurements were taken:

Anthropometry and muscle thickness: Height and body mass were measured with a stadiometer and measuring scales (Model 770; InBody Corporation, Seoul, South Korea). Participants fasted for 12 h before testing, avoided alcohol for 24 h, and voided their bladder immediately before testing.

Muscle thickness (MT) was assessed according to the procedure described by Coleman et al. (2024). The reliability and validity of ultrasound in determining MT has been reported to be very high when compared to the “gold standard” magnetic resonance imaging (Stokes et al., 2021). The same trained ultrasound technician performed all testing using a B-mode ultrasound imaging unit (Model E1; SonoScape, Co., Ltd., Shenzhen, China). The technician applied a water-soluble transmission gel (Aquasonic 100 Ultrasound Transmission gel, Parker Laboratories Inc., Fairfield, NJ, USA) to each measurement site, and a 4–12 MHz linear array ultrasound probe was placed along the tissue interface without depressing the skin.

For the elbow flexors, assessments were conducted on the anterior surface of the upper arm at 45% and 55% of the distance between the antecubital fossa and the acromion process. For the elbow extensors, assessments were obtained on the posterior surface of the upper arm at 45% and 55% between the olecranon tip and the acromion process. When the quality of the image was deemed satisfactory, the technician saved the image to a hard drive and obtained MT dimensions by measuring the distance from the subcutaneous adipose tissue-muscle interface to the muscle-bone interface. Images were obtained at least 72 h post-training to minimize the potential effect of acute muscle swelling. Three images were averaged for each site to derive the final MT value. The test-retest intraclass correlation coefficient (ICC) from our lab for MT measurements is excellent (>0.94), with coefficients of variation (CV) ≤3.3%.

Dynamic muscle strength-endurance

Dynamic upper-body strength-endurance was assessed via 10 RM testing in each respective ROM both pre- and post-intervention for the unilateral lat pulldown exercise. Repetition maximum testing was consistent with recognized guidelines as established by the National Strength and Conditioning Association (NSCA, 2016). In brief, following ROM instruction, participants performed a five-repetition warm-up set of the exercise at ~50% estimated 10 RM, followed by one or two sets of 2–3 repetitions at a load corresponding to ~60–80% estimated 10 RM. Participants then performed a set of 10 repetitions at a heavier load. If successful, they attempted a heavier load for 10 repetitions, continuing until they failed to complete 10 repetitions. Weights attempted were multiples of 5 lbs. When necessary (i.e., if the difference between the last successful attempt and a failed attempt was greater than 5 lbs.), the weight was reduced and another attempt was granted in order to accurately gauge the 10 RM. The heaviest successful attempt was recorded as their 10 RM. One minute of rest was provided between warm-up sets, and 3 to 5 min of rest were provided between each successive 10 RM attempt.

Blinding

To reduce potential bias, the technician obtaining MT measurements was blinded to condition allocation and all statistical analyses were performed by a blinded statistician.

Statistical analysis

All analyses were conducted in R (version 4.4.0) within a Bayesian framework (van de Schoot et al., 2021). Bayesian statistics represents an approach to data analysis and parameter estimation based on Bayes’ theorem, offering several advantages over frequentist approaches, including the formal inclusion of information regarding likely differences between intervention conditions based on knowledge from previous studies (e.g., through informative priors (Swinton & Murphy, 2022)) and the presentation of inferences based on intuitive probabilities (Magezi, 2015).

We used a modified version of the workflow suggested by Gelfand & Wang (2002) to estimate the likely precision of our average treatment effect based on the width of the 95% credible intervals across various potential sample sizes. Additionally, we performed simulation-based calibration of Bayes factors to evaluate whether the correct hypothesis would likely be supported given the sample size and study design (Schad et al., 2023). To assess the expected precision, we first simulated prior predictive data for different sample sizes using informative priors. These priors were derived from a meta-analysis on the topic (Wolf et al., 2023) and from meta-analyses examining the distribution of effects in strength and conditioning (Swinton et al., 2022; Swinton & Murphy, 2022). The priors, set on a standardized scale, included distributions for typical improvement (N(0.44, 0.402)), average treatment effect (N(0.30, 0.272)), heterogeneous response (N(0, 0.152)), and measurement error (N(0, 0.202)). The fitting priors used an average treatment effect prior of N(0, 0.402).

For the simulation-based calibration of Bayes factors, we assumed equal prior probabilities for the null (H0) and alternative (H1) hypotheses. The simulations and model fitting were conducted using a neutral prior of N(0, 0.402), with the average treatment effect set to zero in half of the iterations. Calibration was assessed by examining the average posterior model probability (to determine if it matched the expected 50%) and the average percentage of posterior allocated to the true hypothesis. Models were fit across 500 iterations for sample sizes of n = 20, 25, and 30 (Wolf & Androulakis-Korakakis, 2024) and judged to provide appropriate precision and assessment of strength of evidence of twenty to twenty-five participants.

Inferences were not drawn on within-condition change, as this was not the focus of our research question, although within-condition changes were descriptively presented to help contextualize our findings. The effect of condition (fROM vs. pROM) on outcome variables was estimated using linear mixed models with random effect structures included to account for the within-participant design (Bürkner, 2018).

All inferences were made from posterior distributions of model parameters describing estimates of the effect of intervention allocation and the strength of evidence for either the null or alternative hypothesis of a mean group difference through Bayes factors with a standard scale used to qualitatively interpret the numerical value (e.g., “anecdotal”, “moderate”, “strong” support) (Michael & Wagenmakers, 2014). Informative prior distributions were used based on meta-analysis data on the specific research question and general strength and conditioning literature (Swinton & Murphy, 2022). All analyses were performed using the R wrapper package Bayesian Regression Models (brms) interfaced with Stan to perform sampling (Bürkner, 2018). A complete Bayesian workflow was adopted, which included prior predictive checks, posterior predictive checks, and simulation-based calibration of Bayes factors (Schad et al., 2023). To improve accuracy, transparency, and replication of the analyses, the WAMBS-checklist (When to Worry and how to Avoid Misuse of Bayesian Statistics) was used and reported (Depaoli & van de Schoot, 2017).

Results

Five participants dropped out over the course of the study, resulting in a final sample of 25 participants (training experience = 4.9 ± 4.1 years) that completed the training intervention and pre-/post-intervention testing (see Table 3 for descriptive characteristics).

Table 3 Descriptive characteristics of the participants.

Variable	Men (n = 19)	Women (n = 6)	
Height (cm)	173.9 ± 6.5	164.3 ± 7.5	
Body mass (kg)	80.1 ± 12.1	63.1 ± 7.7	
Age (years)	23.1 ± 4.3	26.2 ± 7.1	

The reasons for attrition were as follows: scheduling issues (n = 2), commute time issues (n = 1), injury unrelated to the study (n = 1), and failure to attend the required number of training sessions (n = 1). Based on the a priori sample size determination, this sample size of 25 participants was judged to provide appropriate precision and assessment of strength of evidence. All participants included in the data analysis completed at least 14 out of 16 (87.5%) possible training sessions. On average, participants completed 96.5% of training sessions.

Muscular adaptations

Initial analyses of within intervention change across outcomes are presented in Fig. 1, with results showing both interventions tending to produce small to medium changes based on thresholds specific to strength and conditioning. Estimates of mean group differences are presented in Table 4. Results were consistent across outcomes with point estimates close to zero and Bayes factors (0.16 to 0.39) in general providing “moderate” support for the null hypothesis of equal improvement across interventions (Table 4). Completion of the WAMBS-checklist identified no issues of concern with the analyses, and nutritional intake appeared to remain relatively consistent across the intervention. These findings suggest that both the lengthened partial and full range of motion interventions led to comparable improvements in muscle hypertrophy and strength-endurance. The similarity in outcomes between the two approaches indicates that neither method provided a clear advantage over the other for enhancing muscular adaptations in this trained population. Pre-post hypertrophy and strength-endurance measurements can be found in Table 5.

Figure 1 Comparative distribution plot of the estimated standardized mean difference of interventions across outcomes.

Density plots illustrate estimates and uncertainty of standardized mean difference changes across the two interventions. Thresholds describing the magnitude of improvements are obtained from strength and conditioning-specific data. N.B. for the elbow flexor 55% measurement, and the standardized mean difference for fROM and pROM are visually indistinguishable (i.e., have near-perfect overlap).

Table 4 Estimated group differences from Bayesian linear mixed models with informative neutral priors.

Outcome	Estimated group difference
[95%CrI]	Posterior
probability	Bayes factor	
Muscle thickness	Probability favoring full range of motion	
Elbow flexor 45%
humeral length (mm)	−0.23 [−1.4 to 0.94]	p = 0.343	0.19: “Moderate” Evidence support of H0	
Elbow flexor 55%
humeral length (mm)	−0.08 [−1.1 to 0.90]	p = 0.438	0.16: “Moderate” Evidence support of H0	
Elbow extensor 45%
humeral length (mm)	0.40 [−1.1 to 1.9]	p = 0.701	0.20: “Moderate” Evidence support of H0	
Elbow extensor 55%
humeral length (mm)	0.82 [−0.44 to 2.1]	p = 0.899	0.39: “Anecdotal” Evidence support of H0	
Strength-endurance	Probability favoring full range of motion	
10 RM full (kg)	−1.2 [−3.7 to 1.3]	p = 0.177	0.30: “Moderate” Evidence support of H0	
10 RM partial (kg)	−0.79 [−3.9 to 2.3]	p = 0.307	0.23: “Moderate” Evidence support of H0	
Note:

Group differences: Positive values favor full range of motion intervention. p-values are calculated from the posterior distribution of the mean group difference parameter and express the probability of a positive value. Bayes Factor: Values less than 1 provide support for the null hypothesis. Values greater than 1 provide support for alternative hypothesis.

Table 5 Pre-post hypertrophy & strength-endurance measurements.

Hypertrophy	pROM	fROM	
Measurement (mm)	Pre-study	Post-study	Pre-study	Post-study	
Elbow flexor 55%	39.6 ± 8.6	41.4 ± 7.8	39.3 ± 7.9	41.1 ± 7.6	
Elbow flexor 45%	36.5 ± 8.9	37.9 ± 8.5	36.5 ± 8.4	37.7 ± 8.5	
Elbow extensor 55%	35.1 ± 9.5	37.0 ± 9.7	34.4 ± 9.6	37.4 ± 9.5	
Elbow extensor 45%	41.5 ± 10.5	44.1 ± 10.3	41.3 ± 10.8	44.4 ± 10.5	
Strength-endurance					
Measurement (kg)	Pre-study	Post-study	Pre-study	Post-study	
fROM 10 RM	63.4 ± 17.1	66.6 ± 16.9	63.6 ± 12.3	65.6 ± 16.5	
pROM 10 RM	66.8 ± 20.7	74.6 ± 21.3	65.4 ± 19.2	72.8 ± 21.6	

Discussion

The current study presents several meaningful findings that provide insight regarding the efficacy of LPs and full ROM RT for stimulating muscular adaptations in resistance-trained individuals. Muscle hypertrophy was similar between conditions, with Bayesian analyses providing anecdotal to moderate support for the null hypothesis (i.e., no difference in effectiveness of either intervention over the other). Similarly, both ROMs appeared to stimulate similar strength-endurance improvements in both partial and full ROM lat pulldown as assessed by 10 RM testing. Herein, we discuss the practical implications of these findings, when considered alongside and compared with findings of the current literature examining the effects of LPs on muscular adaptations.

For muscle hypertrophy, analyses showed moderate evidence in support of the null hypothesis across sites, with the exception of the elbow extensor triceps brachii 55% site, which showed anecdotal evidence in support of the null hypothesis. In all cases, central estimates of group differences were close to zero. There are a number of important considerations when interpreting this study’s muscle hypertrophy results. First, this study represents the most ecologically valid comparison of LPs and full ROM RT to date. Since it compared the two approaches in a multi-exercise, multi-modality RT routine in a resistance-trained population, this study’s results have the greatest likelihood of generalizing to RT practices for muscle hypertrophy in this population. The results obtained here tentatively suggest that LPs and full ROM RT provide similar and effective stimuli for muscle hypertrophy of the elbow flexors and extensors. This should be encouraging for practitioners, as these novel findings allow for considerable flexibility in exercise technique prescriptions. For example, if an experienced trainee is unable to perform a full ROM, or prefers to use pROM, the present evidence suggests the effectiveness of LP ROM RT is similar.

Previous studies, however, comparing LPs and full ROM RT have presented different findings. Briefly, of four studies comparing LPs to full ROM RT, three reported greater muscle hypertrophy from LPs, whereas one found similar muscle hypertrophy (Goto et al., 2019; Kassiano et al., 2022b; Pedrosa et al., 2022; Werkhausen et al., 2021). Moreover, a recent meta-analysis reported a small, potential benefit of LPs over full ROM RT for stimulating muscle hypertrophy (Wolf et al., 2023). While the other studies comparing LPs vs. full ROM RT were conducted in less well-trained populations than the present study, no compelling mechanistic or longitudinal training evidence would suggest divergent adaptations in these populations (Wolf et al., 2024)2 . While acknowledging that more evidence in trained populations is warranted, the totality of available evidence suggests that the effectiveness of LP ROM RT is equal to full ROM RT.

Another important consideration when interpreting the results of this study relates to the muscle lengths being utilized. Previous studies have primarily compared relatively short-muscle length training to relatively long-muscle length training. For example, in a study conducted by Pedrosa et al. (2022), the LP group trained through 100–65 degrees of knee flexion. While greater joint angles are not linearly associated with greater motor tendon unit length (Raiteri, Beller & Hahn, 2021), these joint angles suggest that the quadriceps femoris muscle was not being trained through its longest possible muscle lengths. Indeed, most trainees are capable of approximately 140–150 degrees of knee flexion (Kubo, Ikebukuro & Yata, 2019; Straub & Powers, 2024). With the exception of a study in the gastrocnemius by Kassiano et al. (2022a), wherein the muscle length achieved was likely near maximal, other studies to date compared modestly shorter-muscle length training to modestly longer-muscle length training.

In contrast, the present study represents a comparison of full ROM RT and lengthened partial RT, both with an emphasis on the lengthened position. In both conditions, research assistants ensured participants were reaching the longest-muscle lengths achievable during the exercise. Additionally, both conditions employed a brief pause in the fully stretched position to accentuate the effect. As a result, the fROM condition in the present study also emphasized the stretched position. In the pROM condition, the average muscle length utilized was even greater; however, both conditions involved training at long-muscle lengths (see Fig. 2).

Figure 2 Example of the difference in ROM between the fROM and pROM conditions.

Since similar muscle hypertrophy was observed between conditions, we posit that there may be a point of diminishing - or ceasing - returns to longer-muscle length training, such that training at maximal or near-maximal muscle lengths may not be more beneficial than simply training at sufficiently long-muscle lengths. Indeed, the fROM condition observed similar muscle hypertrophy as the pROM condition in the present study, in contrast to previous studies that found a hypertrophic benefit to training at longer-rather than shorter-muscle lengths. Additionally, it appears that the inclusion of shorter-muscle length training by the full ROM condition did not enhance muscle hypertrophy, suggesting that the inclusion of the lengthened range of motion should be the primary consideration when it comes to range of motion during RT for muscle hypertrophy. This hypothesis is consistent with much of the previous research on the topic, showing a hypertrophic superiority of training at longer vs. shorter muscle lengths (Akagi, Hinks & Power, 2020; Alegre et al., 2014; Bloomquist et al., 2013; Burke et al., 2024; Goto et al., 2019; Hinks et al., 2021; Kassiano et al., 2022a; Kinoshita et al., 2023; Kubo et al., 2006; Kubo, Ikebukuro & Yata, 2019; Larsen et al., 2024a, 2024b; Maeo et al., 2020, 2022; McMahon et al., 2014a, 2014b; Noorkõiv, Nosaka & Blazevich, 2014; Pedrosa et al., 2022, 2023; Sato et al., 2021; Valamatos et al., 2018).

Importantly, the similar magnitude of muscle hypertrophy observed may be a result of the multi-exercise, multi-modality approach employed in the present study. Previous comparisons of LPs and full ROM RT have exclusively used single-exercise interventions (Goto et al., 2019; Kassiano et al., 2022a; Pedrosa et al., 2022; Werkhausen et al., 2021), which appear to limit the homogeneity of muscle hypertrophy observed across a muscle’s different regions (Kassiano et al., 2022b). This may explain the divergence between previous studies and the current study’s results; in the current study, four different exercises targeting each of the assessed muscle groups (eight exercises in total) were employed within each training week.

In terms of muscle strength-endurance, analysis revealed moderate evidence of no difference between LPs and full ROM for increasing both full and partial ROM lat pulldown 10 RMs. This contrasts with several studies that have demonstrated ROM-specific strength acquisition, where training within a specific ROM led to greater performance improvements within that same ROM (Bloomquist et al., 2013; Kubo, Ikebukuro & Yata, 2019; Martínez-Cava et al., 2022; Massey et al., 2005). Notably, these prior studies defined partial-ROM as shorter muscle lengths (e.g., top-down partials in squat or bench press), while our study’s partial-ROM was performed at longer muscle lengths. It is therefore possible, though speculative, that partial-ROM training could provide better transference to full-ROM strength if the partials are completed at longer muscle lengths. Additionally, the results in this study may have been influenced by the cross-education effect, where training one limb with a specific ROM promotes adaptations in the contralateral limb (Bell et al., 2023). Nevertheless, several previous studies have observed strength-endurance improvements across full and partial ROM in resistance-trained populations, regardless of the ROM used during training. While specificity appears to play a small but significant role in maximal performance improvements, our findings suggest that the effect of ROM-specificity may be less influential than anticipated (Crocker, 2000; Hartmann et al., 2012; Rhea et al., 2016). This may also relate to exercise choice, as the lat pulldown is a relatively simple movement where motor learning specificity may be less critical (Rossi et al., 2018).

The present study is not without limitations, which should be considered when drawing evidence-based conclusions from its findings. First, it is possible that the study recruited too small a sample to detect appreciable changes, given sample size determination was based on assumed differences obtained from a meta-analysis of previous studies with a different training status to the participants in the present study. Second, the duration of the study was only 8 weeks. This duration may not have been a sufficiently long timeframe to produce meaningful hypertrophic differences between conditions that could be detected with the sample size. Duration of the study may be particularly relevant in a well-trained population. It is also possible that some of the exercises used were novel to many of the participants (i.e., single-arm bayesian curl, dumbbell overhead extension). The novelty of the exercises may have reduced the hypertrophy observed (Gabriel, Kamen & Frost, 2006). Whilst the results presented here show that the duration was sufficient to induce small to medium improvements, differential adaptations may require a longer interventional period. Third, the muscle strength-endurance findings should be cautiously interpreted, as the cross-education effect between limbs may have confounded results given the within-participant, contralateral limb comparison design. Similarly, since participants always began testing with fROM strength-endurance testing followed by pROM strength-endurance testing, the true change in pROM strength may have been confounded by the fatigue caused by the preceding fROM 10 RM test. This may explain the divergence in results between the present study’s results and the literature on the specificity of ROM at-large (Wolf et al., 2023). Fourth, the present study only measured muscle hypertrophy of the elbow flexors and extensors. Although previous studies have examined other muscles including the ankle plantar flexors, knee extensors, and hip extensors, it is unclear whether these results would generalise to all muscle groups. Another limitation is that, due to the ecological validity of the study’s training protocol, the ROM was not strictly standardized or precisely measured across subjects and conditions. While this enhances the study’s external validity, it reduces its internal validity. Future research in trained individuals should focus on highly standardized ROM protocols to address this issue. Additonally, strength-endurance was assessed only in the lat pulldown; we thus cannot necessarily extrapolate these findings to other exercises, particularly those with different strength curves. Finally, failure for the full ROM condition was defined as the point where participants could no longer complete another full ROM repetition despite attempting to do so, whereas for the partial ROM condition, failure occurred when another partial repetition was no longer possible despite similar effort. This distinction highlights that the two conditions had somewhat different failure points, which may have influenced the results.

Conclusion

The present study showed that LPs and full ROM RT stimulated similar increases in MT of the elbow flexors and extensors over 8 weeks of RT in resistance-trained participants. As the first study in resistance-trained participants, as well as the first to employ a multi-modality, multi-exercise RT intervention, these findings have important implications for the experienced trainee seeking to maximize muscle hypertrophy. Based on the results of this study, alongside others, there appears to be a benefit of emphasizing the lengthened position, whether by use of a full ROM with an emphasis on the lengthened position or LPs. The addition of shorter-muscle length ROM in the present study did not appear to enhance muscle hypertrophy compared to exclusively using LPs, calling into question its role in eliciting increases in muscle size. For muscle strength-endurance, both a full ROM and LP RT stimulated similar improvements in both full ROM and partial ROM muscle strength-endurance; however, other studies have suggested that performance adaptations are muscle-specific. Given the totality of current evidence, it appears prudent to train in the specific range of motion of the desired performance adaptation.

The authors would like to extend their gratitude to the following research assistants, without whom this study would not have been possible: Maxwell A. Kim, Anthony Carrano, and Chris Parnell.

Additional Information and Declarations

Competing Interests

Brad J. Schoenfeld previously served on the scientific advisory board for Tonal Corporation, a manufacturer of fitness equipment. Jeff Nippard is the director at STRCNG Incorporated o/a Jeff Nippard Fitness. The other authors declare no conflicts of interest.

Author Contributions

Milo Wolf conceived and designed the experiments, performed the experiments, prepared figures and/or tables, authored or reviewed drafts of the article, and approved the final draft.

Patroklos Androulakis Korakakis conceived and designed the experiments, performed the experiments, prepared figures and/or tables, authored or reviewed drafts of the article, and approved the final draft.

Alec Piñero conceived and designed the experiments, performed the experiments, prepared figures and/or tables, authored or reviewed drafts of the article, and approved the final draft.

Adam E. Mohan performed the experiments, authored or reviewed drafts of the article, and approved the final draft.

Tom Hermann performed the experiments, authored or reviewed drafts of the article, and approved the final draft.

Francesca Augustin performed the experiments, authored or reviewed drafts of the article, and approved the final draft.

Max Sapuppo performed the experiments, authored or reviewed drafts of the article, and approved the final draft.

Brian Lin performed the experiments, authored or reviewed drafts of the article, and approved the final draft.

Max Coleman conceived and designed the experiments, performed the experiments, authored or reviewed drafts of the article, and approved the final draft.

Ryan Burke conceived and designed the experiments, performed the experiments, authored or reviewed drafts of the article, and approved the final draft.

Jeff Nippard conceived and designed the experiments, authored or reviewed drafts of the article, and approved the final draft.

Paul A. Swinton conceived and designed the experiments, analyzed the data, prepared figures and/or tables, authored or reviewed drafts of the article, and approved the final draft.

Brad J. Schoenfeld conceived and designed the experiments, performed the experiments, prepared figures and/or tables, authored or reviewed drafts of the article, and approved the final draft.

Human Ethics

The following information was supplied relating to ethical approvals (i.e., approving body and any reference numbers):

The Lehman College, Department of Exercise Science and Recreation approved this study (2024-0218).

Data Availability

The following information was supplied regarding data availability:

The data and anonymized videos of participants performing resistance training sessions in both conditions are available at OSF: Wolf, Milo, and Patroklos Androulakis-Korakakis. 2024. “Full Range of Motion vs. Partial Range of Motion at Longer-Muscle Lengths: Effects on Upper-Body Muscle Morphology and Strength.” OSF. September 30. doi:10.17605/OSF.IO/A6CPZ.

1 The studies by Larsen et al. (2024a, 2024b) are pre-printed and have yet to undergo peer-review.

2 This systematic review is currently under peer-review.

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
