# Peer review of "Lengthened partial repetitions elicit similar muscular adaptations as full range of motion repetitions during resistance training in trained individuals"

_PeerJ, doi:10.7717/peerj.18904_

## Round 0.1 · original submission · Major Revisions

Both reviewers have provided extensive comments. Please respond in detail to all them.

Additionally, Provide more context for strength-endurance in the introduction.

·

Basic reporting

A relevant and interesting study for resistance training populations that aims at maximizing resistance training-induced muscle hypertrophy. The manuscript is mostly written in a clear language with a clear aim and direction. In addition, I applaud the authors' effort for making data and data collection transparent to a degree where even video of the training instruction is made available.
I do however have some major inputs on how data should be reported. In particular, I strongly urge the authors to rewrite the results section and provide a more clear visualization of the results. This would greatly improve the manuscript. Besides this I have listed a point-by-point summary in additional comments. All of these points should be addressed, as it would strongly benefit the manuscript.

Experimental design

The aim of the study is clearly defined and relevant for the scope of the journal. It is also valuable data for resistance training populations. While I have comments to specific parts of the writing (see point-by-point summary in additional comments), the study is adequately designed to answer the research question. While there are some limitations, most of these have been addressed by the authors, and addressing my specific comments will further ensure that all limitations are addressed appropriately. I believe the authors have done everything they could to ensure as high a technical standard for the present study given their circumstances (i.e. equipment availability etc.). All aspects of the study are described in a manner that would allow for replication, although parts could be made more clear and concise (see comments).

Validity of the findings

All underlying data has been provided and a robust statistical approach has been used with an appropriate interpretation. Both the discussion and conclusion are written in a mostly clear manner. I enjoyed these parts of the paper and only have minor comments here.

Additional comments

Abstract:
- The abstract could benefit from a one sentence background before the purpose. Just to introduce the topic to the reader.
- Be more concise in the methods. Fx no need to write “The RT-intervention was a multi-exercise, multi-modality eight-week program”. Just describe the training that was done as concisely as possible.
- No need to abbreviate resistance training, abbreviations should only be used if they aid the reading process.
- In the conclusion section, stick to what you can conclude from your data. This is not a review, so save incorporating other studies results for the discussion. Thus, the conclusion here is not in line with your findings. This needs to be fixed.

Introduction:
- You spend the entire first paragraph of the introduction discussing mechanisms, but do not investigate any mechanisms in the paper. This needs to be shortened substantially and the subject of ROM introduced earlier instead.
- Not a fan of the abbreviations such as LPs and SPs. I would suggest finding a way to work around this in the writing without abbreviating. This would improve the reading experience, but is not crucial.

Methods:
- I do not understand the approach taken to recruiting participants. You should be more specific here. While a power calculation was not made (probably since you opted to use a Bayesian statistical approach), you can still reference other studies findings to justify the sample size. As it is written now it is very vague.
- Save describing the Bayesian framework for the statistics section. Also as it pertains to determination of sample size. It does however need to be a lot more concise than this. You should be able to describe Lines 81-108 precisely in only a few lines.
- You argue that some of the novelty in this study lies in that previous studies used untrained or recreationally trained, and that you therefore aim at a more trained population. Despite this you use the criteria of at least one upper body resistance training session per week over the past 6 months. This could easily include subjects only being recreationally trained. You need to change your text accordingly.
- Line 121-122: I do not understand this. Did you stick to the inclusion criteria, and included all volunteers that met these criteria, or did you have a biased approach by selecting specific participants from a larger number of volunteers fx to “emphasize those with greater training experience”?
- Line 126-143: Should be its own section, called Study design, or something similar
- Why did you assess 10RM unilateral lat pulldown when you only assess upper arm muscle thickness? (i.e. you assess muscle hypertrophy of upper arm, but test performance only in one multi-joint exercise)
- The full ROM condition terminates a set when they cannot complete another full rep, the partial ROM when they cannot complete 50% of the repetition. Thus, by definition, the failure point for these conditions are different, with the partial ROM group technically training harder (i.e. failure at half reps vs. full reps). I understand why this was done as you opted for within participant design and the only way to control for this would be a parallel group design with a third group. However, this should still be noted as a limitation of the study in the discussion.
- Line 202: I guess you forgot to put a reference here?
- Line 237-243: This is unnecessary. It is fine if you want to argue for your statistical approach but this should be done very short and concise. Ideally not more than a single sentence.
- Line 251: Needs a “.” After the reference.

Results:
- Make it more clear that 30 participants participated in the study before you mention that five participants dropped out. Currently the only place to see that 30 participants were recruited is in the parentheses with “n=30” in the methods section. At no point is it mentioned explicitly.
- Only 6 women were included vs 19 men. This is not inherently a problem for the research question at hand, but it contradicts your previous statement that you prioritized recruiting women. Therefore the previous statement should be removed.
- Line 282-289: Major rewrite required here. Explain the results, they should be understandable even without looking at the figures. This is not the case currently.
- Table 4 and 5 should be combined for better overview of results.
- I suggest separating figure 1 into different panels for the different findings and finding a more clear way of visualizing the changes. Fx scatter plot of individual changes and mean with 95%CI. In the same vein, no reason to use standardized mean difference. By doing the suggested, just use the actual outcome values.


Discussion:
- Line 300-301: Rewrite more conservatively. Several findings is quite a stretch, your finding is that long length partials and full ROM seems to elicit similar hypertrophy and strength in the context of the present study.
- Line 309-312: Avoid statistical jargon in the discussion and discuss the interpretation of the results in a clear and concise manner instead.
- Line 381-383: Again, this can be written more clearly than just restating the results from the statistical analysis.
- Line 404: I agree, but elaborate on why this is the case (i.e. why duration is particularly relevant in a well-trained population).

Conclusion:
- Line 437-439: It is possible, but it is not something that this data indicates, so keep this in the discussion but remove it from conclusion.

·

Basic reporting

Clear, unambiguous, professional English language used throughout.
Reviewer comment: Yes, the language is clear and unambiguous throughout.

Introduction and background to show context. Literature is well referenced and relevant.
Reviewer comment: The introduction is well-written and provides sufficient context for hypertrophy outcomes. In my general comments below, I ask the authors to provide some context for strength-endurance outcomes.

Structure conforms to Peer J standards, discipline norm, or improved for clarity.
Reviewer comment: Yes.

Figures are relevant, high quality, well-labelled, and described.
Reviewer comment: The figures are relevant and high quality. In my general comments below, I ask the authors to provide more information for some of the figures.

Raw data supplied.
Reviewer comment: I was not provided with raw data from the Peer J portal.

Experimental design

Original primary research within scope of the journal.
Reviewer comment: Yes, this is an original research study that is within the scope of the journal.

Research question well defined, relevant, and meaningful. It is stated how the research fills an identified knowledge gap.
Reviewer comment: Yes, the research question is well defined, relevant, and meaningful. The knowledge gap is clearly defined and the authors address the knowledge gap directly.

Rigorous investigation performed to a high technical and ethical standard.
Reviewer comment: Yes, the study was completed with high technical and ethical standards.

Methods described with sufficient detail and information to replicate.
Reviewer comment: Yes, the methods are clearly described with sufficient detail for replication; however, I do have one small general comment for clarity below.

Validity of the findings

Impact and novelty is assessed. Meaningful replication is encouraged where rationale and benefit to literature is clearly stated.
Reviewer comment: Yes, to all of the above.

All underlying data have been provided; they are robust, statistically sound, and controlled.
Reviewer comment: The data have been provided, are robust, statistically sound, and controlled. In my general comments below, I ask the authors to provide data for control of dietary intake.

Conclusions are well stated, linked to original research questions, and limited to supporting results.
Reviewer comment: Yes, to all of the above.

Additional comments

Abstract
- Line 20: Consider specifying that the 10-RM was assessed via unilateral lat pulldown here and throughout the manuscript.
- Overall the abstract is well-written; concisely and thoroughly summarizes the study.

Introduction
- Line 59: Do the results from Larsen, Swinton et al. stem from training with lengthened partials or the additional volume, and arguably forced repetitions, completed by the lengthened partials group? Consider adding a sentence for further clarity and explanation. In this study, it appears that the group that trained beyond failure completed significantly higher total volume load compared to the group that did not, so the reader may benefit from being provided this context.
- Line 61: Because 3 out of the 4 cited studies concluded that lengthened partials were superior to full-ROM training, I would like the authors to speculate why Werkhausen et al. reported disparate outcomes. Based on their methods section, the subjects in their study performed a concentric-only protocol and the range of motion trained by the partials group was extremely small (i.e., a 9o change in knee angle during a leg press). This may help explain their seemingly unique results.
- Line 75: The authors clearly state that the purpose of the study was to compare the effect of lengthened partials and full-ROM training on upper-body muscle hypertrophy, but there is no mention of strength-endurance in the purpose statement or the introduction. Because strength-endurance (i.e., increases in 10-RM) was measured in this study, and an interesting outcome was observed for this dependent variable, I recommend the authors add background information pertaining to the effect of lengthened partials versus full-ROM training on muscular strength to the introduction and purpose statement. Or, more simply, add the evaluation of the effects of full-ROM and LPs on strength-endurance as a secondary purpose at the end of the introduction.

Methods
- Line 149: Looks like a typo here where ‘weight’ was written instead of ‘eight.’
- Line 154: Consider defining ‘momentary muscular failure’ for readers who are not familiar with this term.
- Line 159: Based on how this sentence is written, it seems like the subjects alternated between their full-ROM and lengthened-partial limb during the training sessions. Did the subjects perform 4 sets of an exercise with one limb before performing 4 sets of the same exercise with the other limb, or did the subjects alternate limbs set-by-set? Please clarify.
- Line 178: Looks like a missing word here. Perhaps add ‘was’ before ‘randomized.’
- Line 202: It appears that a citation is missing here as ‘X’ is listed instead of a reference number.

Results
- Line 269: Looks like a typo here, please add an opening parentheses before ‘training experience.’
- Line 287: Referring to Table 5, it is helpful to see the group means for the hypertrophy data. I recommend the authors add the group means for strength-endurance data to Table 5.
- Line 289: I appreciate that the authors took the time to collect nutritional data from their subjects; however, the statement ‘nutritional intake appeared to remain relatively consistent across the intervention’ may be too opaque for the readers. If the data are available, the manuscript could benefit from a table that displays total energy intake and macronutrient distribution from the pre and post 5-day food logs, as outlined in the methods section.

Discussion
- Line 314: Lengthened partials could be abbreviated as LPs here because it was defined as such in the introduction of the manuscript.
- Line 319: This sentence is a bit wordy; consider deleting ‘a high level of room for’ and replacing it with ‘high’ or ‘considerable’ or some other synonym.
- Line 322: It may be outside of the scope of this manuscript, but I am curious if these findings would be beneficial to athletic trainers and physical therapists who may have patients train at a particular ROM to avoid painful ROM during phases of rehabilitation.
- Line 386: Here, I think the authors have an opportunity to further discuss the interesting finding that 10-RM strength did not differ between LPs and full-ROM. The notion that strength acquisition is ROM-specific has been demonstrated with squat (Bloomquist et al., 2013; Kubo et al., 2019) and bench press (Martinez-Cava et al., 2022; Massey et al., 2005) interventions. Collectively, these studies reported inferior full-ROM strength development when partial-ROM interventions were applied. However, in these studies, regardless of how partial-ROM was defined (e.g., 60o of knee flexion on a squat; 1/3 of full-ROM on a bench), the partial repetitions were performed top-down, not bottom-up, meaning that repetitions were performed at shorter muscle lengths. In other words, it is unknown if partial-ROM would have increased full-ROM strength if the partials were performed bottom-up at longer muscle lengths. Based on your data, and although it is speculative in nature, it is possible that partial-ROM provides better transference to full-ROM strength provided that the partials are completed at long muscle lengths. Although hypertrophy is the primary focus of this study, the strength data are intriguing and may fill another gap in the resistance training literature. I encourage the authors to expand the discussion here.
o Bloomquist et al., 2013: https://pubmed.ncbi.nlm.nih.gov/23604798/
o Kubo et al., 2019: https://pubmed.ncbi.nlm.nih.gov/31230110/
o Martinez-Cava et al., 2022: https://pubmed.ncbi.nlm.nih.gov/31567719/
o Massey et al., 2005: https://pubmed.ncbi.nlm.nih.gov/15903383/
- Line 390: I would like the authors to provide citations after this sentence, especially because it is prefaced by ‘several previous studies have observed…’.
- Line 416: There is a call to Figure 8 at the end of this sentence, but the manuscript only contains three figures. Is this a typo or are the authors calling to a Figure 8 in the manuscript by Wolf et al. 2023?
- A final comment about the lack of difference in strength-endurance acquisition between the two interventions: did the authors record the external loads (e.g., 60 lbs.) lifted with full-ROM and LPs or perhaps total volume load accrued during the training sessions? Based on your methods section, it seems that loads were adjusted set-by-set to allow the lifter to achieve momentary failure within a particular repetition range, but did the LP limbs use heavier external loads to elicit failure in the same repetition range as the full-ROM limb? Or was external load matched between the two arms? I understand that the resistance training literature has evolved in recent years, and there is now evidence that muscular strength and endurance can be increased with a variety of relative intensities and corresponding repetition ranges, but could a potential difference in external load explain why the LP intervention increased full-ROM strength to a similar degree as the full-ROM intervention?
Figure 2: This is a great figure that clearly explains why the present study truly explored LPs more so than previous studies. Consider providing a foot note to identify what ‘Study 1’ and ‘Study 2’ are.
Figure 3: Another great figure that helps the reader visualize how fROM and pROM were executed in this study. However, the label is a bit misleading because it reads ‘mean muscle length during resistance training in existing studies.’ First, the images provided seem to depict how fROM and pROM were performed in the present manuscript, not others. Second, Figure 2 helps the reader identify differences between ROM in ‘existing studies’ versus the present study, which causes Figures 2 and 3 to seemingly contradict each other. This figure is called to after the sentence in line 354, where the authors appear to be bringing attention to the ROM applied in this present study, which distinguishes it from existing studies. Please clarify.

·

Basic reporting

Relevant prior literature is appropriately referenced and provided; however, a sufficient field background/context is not provided. I recommend writing a paragraph in the Introduction that provides a short background on the effect of resistance training on strength-endurance. Also, I recommend adding to your purpose statement that you compared the effect of LPs and full ROM RT on hypertrophy and strength-endurance. You could also just state that you compared the effect of LPs and full ROM RT on muscular adaptations. See the comments I made within the annotated PDF for further details.

Experimental design

The methods section is mostly described with sufficient detail and information to replicate. However, I recommend adding a section at the beginning of your methods that provides an overview of your study and experimental design. Also, a few areas of the Resistance Training Procedures, Assessments, Tables, and Figures need clarification. See the comments I made within the annotated PDF for further details.

Validity of the findings

No comment.

Additional comments

When I review an article, I like to provide as much constructive feedback as possible to help improve the manuscript. I also like to bring to the authors’ attention grammatical errors that I catch when reading the manuscript. See the annotated PDF for further details.

---

## Round 0.2 · Minor Revisions

The reviewers are very happy with revisions but just some minor changes are needed.

·

Basic reporting

The authors have accommodated most of my comments and meet the requirements for basic reporting. See additional comments for minor changes that should be incorporated before publication.

Experimental design

My stance remains the same as during first revision, the aim of the study is clearly defined and relevant for the scope of the journal, and the study is adequately designed to answer the research question.

Validity of the findings

My stance remains the same as during first revision. All underlying data has been provided and a robust statistical approach has been used with an appropriate interpretation. I only have a minor comment to presentation of data (see additional comments).

Additional comments

Abstract
No further comments

Introduction
- Line 37-40 is irrelevant to the manuscript. You should just go straight from line 37 to line 42.

Methods
No further comments

Results
- I do not understand why no attempt to change the figure was made. The manuscript could greatly benefit from better visualization of data. Only 1 figure of the data is presented, and this could be substantially improved by using more intuitive visualization as suggested previously. In particular, I do not understand why the authors persist on sticking with standardized mean differences with arbitrary cut offs for effect sizes when presenting the absolute changes would be much more informative.

Discussion
- Line 39: You have by mistake written “LPs” twice here.
- You write in your response to reviewers that you have incorporated the limitation of different failure points in the full ROM vs partial ROM condition. However this is not the case. Needs to be fixed.

·

Basic reporting

No comment.

Experimental design

No comment.

Validity of the findings

No comment.

Additional comments

I appreciate the work that the authors put into the revised manuscript. My initial comments/concerns were adequately addressed, especially the expansion of the strength-endurance piece in the discussion. This is an interesting study with several practical implications. Below, I have a few minor comments for the authors to consider.
Line 62 – There are two periods at the end of this sentence. Please remove one.
Line 165 – Just a minor detail here. In this sentence, the testing procedure is referred to as “10RM testing” and then “10-RM testing.” I recommend picking one, with or without the hyphen, and then staying consistent throughout the manuscript.
Line 299 – Consider adding “strength-endurance” to this sentence because this information is now provided in Table 5. Thank you for adding these data to the Table.
Lines 397-398 – I appreciate the authors for citing the studies that were recommended to expand the strength-endurance discussion. This paragraph reads very well. Please double-check that all new citations have been added to the reference list. It looks like Martinez-Cava et al. (2022) and Massey et al. (2005) were not added.
Figures 2 and 3 – Looking at these figures again, it seems like their labels may be accidentally flipped. Based on how they are called to within the body of the manuscript, Figure 2 is contrasting the muscle lengths trained in the present study versus previous studies, but it is labeled as “Example of the difference in ROM between the fROM and pROM conditions.” Moreover, Figure 3 intends to demonstrate the fROM and pROM conditions in the present study, but it has been labeled as “Mean muscle length during resistance training in existing studies.” Please clarify.

·

Basic reporting

• The article is written in English and uses clear, unambiguous, technically correct text. The article conforms to professional standards of courtesy and expression.
• The article now includes a sufficient introduction and background to demonstrate how the work fitness into the broader field of knowledge. Relevant prior literature is appropriately referenced.
• The structure of the article conforms to an acceptable format of ‘standard sections.’
• Figures are relevant to the content of the article, of sufficient resolution, and appropriately described and labeled.
• All appropriate raw data have been made available in accordance with PeerJ’s Data Sharing policy.
• The submission is ‘self-contained,’ represents an appropriate ‘unit of publication,’ and it includes results relevant to the hypothesis.
• Coherent bodies of work are not inappropriately subdivided merely to increase publication

Experimental design

• This study is primary and original, and it is within the Aims and Scope of the journal.
• The submission clearly defines the research question, and it is relevant and meaningful. The knowledge gap being investigated is identified, and statements are made as to how the study contributes to filling that gap.
• The investigation was rigorously conducted to a high technical standard. The research was conducted in conformity with the prevailing ethical standards ion the field/
• The methods are described with sufficient detail & information to be reproducible by another investigator.

Validity of the findings

• All underlying data have been provided; they are robust, statistically sound, & controlled.
• Conclusions are well stated, linked to original research question & limited to supporting results.

Additional comments

See below for my response to your comments to my first review. I also included another annotated manuscript with further comments and recommendations. Please address each comment and let me know why you did or did not decide to make the change I suggested in my previous review.

Your attention to detail, from enhancing the title to ensuring consistency in abbreviations, has
truly elevated the overall quality of the manuscript, thank you for your thorough and thoughtful review. Since all of your comments were provided in an annotated PDF, we’ve included an overview-style response to some of them below.
• Thank you for the kind compliments. I am glad that I was able to help you all enhance your title and ensure consistency in abbreviations.

We have updated the title to “Lengthened Partial Repetitions Elicit Similar Muscular
Adaptations as Full Range of Motion Repetitions During Resistance Training in Trained
Individuals” as suggested.
• Thank you.

We have ensured all abbreviations, such as fROM (full range of motion) and pROM
(partial range of motion), are defined clearly and used consistently throughout the
manuscript.
• Thank you taking my feedback and suggestion to improve the clarity and consistency of your manuscript.

We have replaced the term “excursed” with “utilized” for clarity.
• Thank you. I like the word you chose.

We added a brief rationale explaining that varying repetition ranges were included to
maintain ecological validity.
• Thank you.

As advised, we have revised sections for conciseness, ensuring that all content,
particularly in the Methods and Discussion sections, is precise and to the point.
• Thank you.

We have expanded the Discussion to include justifications and comparisons relevant to
the strength-endurance and ROM findings, including references to relevant literature.
• Thank you for these additions. This will strengthen your manuscript.

We have added a table with pre- and post-intervention strength-endurance results as
requested.
• Thank you.

---

## Round 0.3 · Minor Revisions

Please see Reviewer 3 comments and address all comments on the annotated pdf file. Thanks so much.

·

Basic reporting

The article meets requirements for publication. See previous review for specifics. Final comments are listed in "additional comments".

Experimental design

The article meets requirements for publication. See previous review for specifics. Final comments are listed in "additional comments".

Validity of the findings

The article meets requirements for publication. See previous review for specifics. Final comments are listed in "additional comments".

Additional comments

Congratulations on the manuscript. With the current revisions I believe that the manuscript is appropriate for publication. Two minor comments that I encourage fixed:

- Line 39: ”has” should be replaced with “have”

- The argumentation for not improving the figure is weak. As is, tables report absolute values which I agree is useful for transparency. However, this does not justify that changes with the intervention cannot be presented in a more intuitive and visually appealing manner in an improved figure. My comment was not specific to a certain way of presenting the figure, it was just examples of how the figure could be improved. Despite this, no attempt has been made to improve the figure at all. I still encourage that an attempt is made to improve the figure before publication as I still believe it is an underwhelming visualization of the data.

·

Basic reporting

No further comments.

Experimental design

No further comments.

Validity of the findings

No further comments.

Additional comments

The authors addressed all comments/concerns raised from the second round of review. I believe the manuscript is ready for publication. Thank you for the opportunity to review this work.

·

Basic reporting

The Figures are relevant to the content of the article and of sufficient resolution; however Figure 1, Figure 2, and Figure 3 need to be more appropriately described and labeled (see additional comments #7-11 below and within the annotated pdf. for my suggestions).

Experimental design

No comment

Validity of the findings

No comment

Additional comments

Thank you for your apology and for responding to the comments I made within the annotated pdf file. I have uploaded another annotated pdf and have additional comments for you to address before your manuscript can be accepted for publication. Please review this file and do the same as you with your previous response to my comments. I have highlighted below a few of the key points I need you to address.

1. Line 52: You stated that you removed this, but it is still here. Please remove this because you already defined it in the previous paragraph.
2. Line 53: Does this citation refer to the Kassiano, Costa or Kasiano, Nunes reference or both?
3. Line 213: You have two periods here. Please delete one.
4. Line 256: Remove the si.
5. Line 324: Please write out triceps brachii 55% site. You did not define TB as triceps brachii earlier in the manuscript.
6. Line 371: Change LPs to pROM.
7. Figure 1: You replied "Thank you!" to my previous comments, but you did not make the change in this version of the manuscript. Please clarify. I think you should write this as follows: N.B. for the elbow flexor 55% measurement, and the standardized mean difference for fROM and pROM are visually indistinguishable (i.e., have near-perfect overlap).
8. Figure 1: In my previous review, I suggested that you add a hyphen between conditioning and specific. You said that you added it, but it is not in the current version of the manuscript.
9. Figure 2: Was your mean muscle length during the fROM intervention longer than the mean muscle length of the fROM of Study 1 and Study 2? I interpret this point on the figure as a mean muscle length of 75% lengthened. Shouldn't this point be closer to the Study 1 and Study 2 fROM points on the figure?
10. Figure 2: Please provide a legend that identifies what each of the shaded areas indicate (e.g., blue, dark red, and light red filled areas versus blue/orange and red/orange lined areas).
11. Figure 3: Will you please provide two more visuals (similar to the one you provided in this figure for your study) within this figure that show the ROM-differences of the example Study 1 and Study 2 you cited in Figure 2? This will help the reader better understand the differences in mean muscle lengths from your study and the other two example studies form Figure 2. For example, you can have an A, B, and C part of this figure. Part A could be the current figure. Part B can be a visual of fROM and pROM from Example Study #1. Part C can be a visual of fROM and pROM from Example Study #2. Thanks.

---

## Round 0.4 · accepted · Accept

Congratulations and thank you for your diligence with the reviewers comments.

·

Basic reporting

No comment.

Experimental design

No comment

Validity of the findings

No comment

Additional comments

Thank you for allowing me to review your manuscript and provide constructive feedback. I think removing your previous Figure 2 was a great idea. I believe that you have addressed all of my concerns and recommnedations, and your manuscript is ready to be published. I wish you all the best in your future endeavors. Please do not hesitate to recommend me as a reviewer for any of your future manuscripts.

Respectfully,

Jason Beam